# Enantiotopic-group-selective coupling for unified access to carbazole atropisomers as versatile chiral chromophores

Junqiang Wei[1], Zhuoer Wang[2], Pengyao Xing ®[2] ✉ & Ye Zhu ®[1] ✉

Chiral organic chromophores are foundational for advanced optical and electronic devices. Despite the widespread use of *N*-aryl carbazoles in visible-luminescent materials, chiroptical applications of their atropisomers have remained underdeveloped due to the synthetic challenge of achieving remote atroposelectivity necessitated by extended π-systems. Here, we present a unified strategy for the efficient synthesis of enantioenriched N–C and N–N carbazole atropisomers. By integrating $^{13}$C NMR-based ligand parameterization, we achieve enantiotopic-group-selective coupling reactions that simultaneously incorporate tailored π-functionalities and establish axial chirality (up to >99:1 er) using synthetic pathways established in carbazole chemistry. Through covalent modulation and noncovalent complexation, we investigate novel chiroptical functions of carbazole atropisomers, including circular dichroism (CD), circularly polarized luminescence (CPL), charge-transfer CPL (CT-CPL), and circularly polarized thermally activated delayed fluorescence (CP-TADF). By establishing an electrostatic steering strategy for remote atroposelectivity, our work paves the way for integrating multifunctional carbazoles into advanced optical and optoelectronic technologies.

Chiral chromophores confer precise photon control via quantized circular polarization, underpinning advances in quantum computing, chemosensing, and optoelectronics[1]. Chiral organic chromophores built upon visible luminescent, nitrogen-centered stereogenic scaffolds have remained elusive. *N*-aryl carbazoles are prominent components in organic materials, conjugated polymers, metal–organic frameworks, and covalent organic frameworks, representing a vast class of synthetic chemicals (>1 million substructure entries in SciFinder)[2,3]. Establishing stereogenicity directly about the nitrogen core could represent a compelling strategy for designing novel chiral chromophores that inherit the excellent luminescence and conductivity of the privileged carbazole scaffold.

Despite the prevalence of atropisomerism[4–9], N–C axially chiral *N*-aryl carbazoles have only been utilized in a few cases as chiral ligands[10–12] and organocatalysts[11,12], likely due to the lack of general asymmetric synthetic methods beyond the two reports targeting

*N*-naphthyl *mono*−2-substituted carbazoles via asymmetric *N*-naphthylation (Fig. 1A)[11,12]. Given the dominance of cross-coupling in carbazole derivatization, we envisaged that enantiotopic-group-selective coupling[13–20] of dihalocarbazole could offer a modular approach to establishing stereochemistry without changing the synthetic routes or restricting the types of functional groups introduced.

Catalytic desymmetrization of pro-stereogenic biaryls typically relies on the steric biasing by *ortho* substituents of the nonreacting arenes; therefore, the reaction sites are invariably close to the emergent chiral axes (Fig. 1B)[21–33]. To date, desymmetrizing cross-coupling of 2,6-di(pseudo)halo biaryls are limited to C–C atropisomers[34–37]. Notably, Miller[38] and Akiyama[39] achieved desymmetrizing C–O and C–C coupling of 2,6-diOH and 2,6-diMe biaryls, respectively. However, establishing axial chirality through reactions at remote enantiotopic sites has remained a substantial challenge due to diminished stereochemical effects of the further distanced prostereogenic axis.

[1]Department of Chemistry, Faculty of Science, National University of Singapore, 3 Science Drive 3, Singapore, Singapore. [2]School of Chemistry and Chemical Engineering, Shandong University, Jinan, People's Republic of China. ✉e-mail: xingpengyao@sdu.edu.cn; chmzhu@nus.edu.sg

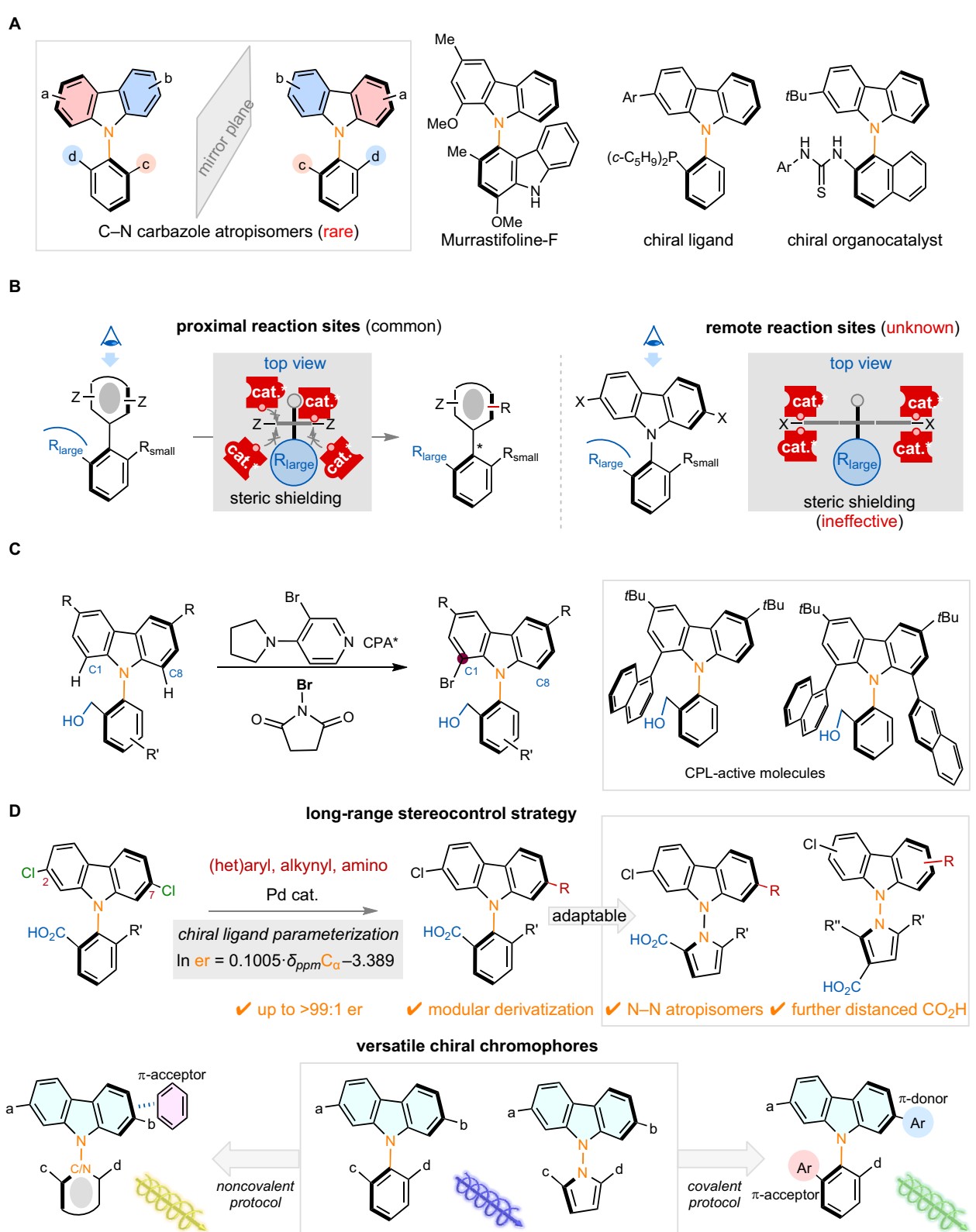

**Fig. 1 | Synthetic strategy for axially chiral carbazoles. A** Atropisomerism of *N*-aryl carbazoles and its applications. **B** Conventional steric biasing does not confer long-range stereochemical relay. **C** Huang, Wong & Yeung's work on organocatalytic desymmetrization to 1-bromo *N*-aryl carbazoles. **D** This work: Catalytic remote desymmetrization offers modular access to carbazole atropisomers for chiroptical functions. Ar aryl, Bu butyl, CD circular dichroism, CPA chiral phosphoric acid, CPL circularly polarized luminescence, CT charge transfer, TADF thermally activated delayed fluorescence.

Specifically, steric biasing of the prochirality-determining substituents ($R^1$ and $R^2$) would become ineffective for the proposed desymmetrizing coupling of prochiral dihalocarbazole. Very recently, Huang, Wong, Yeung and coworkers reported organocatalytic asymmetric bromination at C1 of N-aryl carbazoles using chiral phosphoric acids (CPA) and a Lewis base (3-bromo-4-pyrrolylpyridine, Fig. 1C)[40]. Subsequent cross-coupling furnished novel molecules featuring two or three chiral axes that exhibited circularly polarized luminescence (CPL). Nevertheless, highly efficient synthetic methods to access N−C stereogenic carbazoles with differentiated 2,7-bis-substituents for exploiting of the chiroptical functions of this privileged scaffold remain unexplored.

Herein, we report a unified strategy for accessing axially chiral carbazoles (up to >99:1 er) that overcomes the challenges posed by remote reaction sites, even when the *ortho* groups are not sterically differentiated (Fig. 1D). Unexpectedly, we identified the relationship between atroposelectivity and distal $^{13}$C NMR chemical shifts through ligand parameterization. This strategy was adaptable to various scaffolds, including the previously unexplored atropisomerism of N-aryl phenoxazine and N-aryl dihydroacridine, as well as N−N stereogenic carbazoles—an untapped class of N−N stereogenic compounds[41–49] that lacks enantioselective access[50]. The modular synthesis of N−C and N−N axially chiral carbazoles enabled versatile chiroptical functions, including circular dichroism (CD), circularly polarized luminescence (CPL), charge-transfer CPL (CT-CPL) and circularly polarized thermally activated delayed fluorescence (CP-TADF), thereby setting the foundation for applications in optical and optoelectronic materials.

## Results

### Catalyst development for remote atroposelectivity
We initiated our study by investigating the Suzuki−Miyaura reaction of 2,7-dichlorocarbazole 1 (Fig. 2A). Using 3′-carboxyl L1 as chiral ligand, the reaction proceeded in 17% yield with 70.5:29.5 er. Gratifyingly, stereoinduction was modulated through incorporating an amino acid moiety in the ligand. While ($S_a$, $S$)-L2 was less effective (66:34 er), the reaction using diastereomeric ($S_a$, $R$)-L3 afforded 2 in 83:17 er. Replacing 5′-Me of L1 for 5′-Ph improved the stereoselectivity (L4, 82:18 er). Although introducing an alanine moiety did not make a noticeable difference (L5, 81.5:18.5 er), leucine-derived 5′-Ph ligand gave improved results (L6, 86.5:13.5 er).

Encouraged by the results, we obtained the crystal structures of L6 and 2 to elucidate the stereocontrol (Fig. 2B). The distance between $CO_2H$ and Cl in 2 is 6.8 Å, and the distance between $CO_2H$ and phosphorus atom in L6 is 8.3 Å. It is unlikely that the catalyst differentiates the 2′-$CO_2H$ and 6′-Me of 1 (effective radii 1.62 Å and 1.80 Å, respectively[51]) through steric biasing because of their similar steric properties when compared with their distances from the Cl of 1 (6.8 Å). This was confirmed by the diminished 45:55 er of desymmetrizing reaction when $CO_2H$ of 1 was masked as an ethyl ester.

The chiral ligand's distal carboxylate serves as a nonligating ionic group for substrate recognition in the absence of steric biasing. Based on the absolute configurations of the ligand and the product, we hypothesized a model for stereoinduction. The substrate-catalyst complex is preorganized by the distal ionic interactions between their anionic carboxylates via cation bridges[52,53]. The steric map[54] of L6 indicates that the buried volumes of the four quadrants are very similar ($V_{buried}$ 51.4% to 53.8%).

Therefore, the ligand possesses an electronically asymmetric yet sterically unbiased environment, a feature rarely shared by commonly used chiral catalysts. The crystal structure of L6 reveals hydrogen bonding interactions between 2′-O$i$Pr and amide NH. In addition, the estimated torsion angle H−$C_α$−N−C(O) is 8°, which is consistent with conformations of typical peptides. Compared with the $CO_2H$, the side group (i.e., $i$Bu) points away from the metal center. Such conformation is consistent with the experimental results that the diastereomeric ($S_a$, $S$)-

L2 was less effective than ($S_a$, $R$)-L3. In an analogous conformation, the $CO_2H$ of L2 would be pointing toward the opposite side of the ligand scaffold when viewed from the phosphorus atom.

The tunability of the ligand's amino acid moiety allowed us to improve the stereoselectivity to 96:4 er (L7 to L9, Fig. 2A). Intriguingly, side group R modulates the stereoselectivity, even though it points away from the phosphorus atom. The results contrast with our previous experience that altering the side group R was not influential[55]. Ligand parameterization[56–58] allowed us to further elucidate their effects (Fig. 2B). After surveying common side chain parameters (Table S1 in Supplementary Information, SI), positive correlation between energetic term ln(er) and graph shape index−steric parameter of amino acids−was revealed ($R^2$ = 0.831), but the trend does not hold for L7 and L8. By contrast, we found a strong linear correlation between ln(er) and the experimental $^{13}$C NMR chemical shift (δ ppm) of $C_α$ of the ligands ($R^2$ = 0.992). The results suggest that both the steric bulkiness and the electronic property of R influence the stereocontrol control.

### Reaction development
With the optimal catalyst in place, we explored the reaction scope. A spectrum of (hetero)aryl boronic acids and *p*-anisidine were successfully coupled in high stereoselectivity (up to 99:1 er) (Fig. 3A). Electron-donating (3, 8 and 9) and electron-withdrawing functional groups (4–7) at *para* (3–5), *meta* (6–9) and *ortho* (10) positions were well tolerated. The reaction was applied to heterocycles including pyridine (11), furan (12), thiophene (13 and 14), and extended π-systems commonly used in organic materials (15–20) including dibenzothiophene (18 and 19) and carbazole (20). In addition, Buchwald−Hartwig reaction proceeded in excellent stereoselectivity (21, 99.5:0.5 er), thereby further expanding the functionality of carbazole atropisomers.

Desymmetrization of prochiral biaryls is typically effective for a specific set of *ortho* substituents that preserves the steric biasing by a given catalyst. Considering that ionic stereocontrol could potentially overcome such limitation through electronic differentiation, we probed the ionic catalyst's adaptability to substrates bearing various *ortho* groups (Fig. 3B).

Although 2′-aryl substituted N-aryl carbazoles are widely used as organic materials (>146,000 substructure entries in SciFinder), their enantioenriched forms have remained unknown. Using the same catalyst optimized for model substrate 1, the stereocontrol was effective irrespective of the electronic property of the 6′-aryl (22 vs 23). In addition, incorporating a bulky 6′-(1-naphthyl) resulted in slow rotation along the C−C bond connecting the naphthyl, as revealed in the NMR spectra of 24. Such steric effect does not influence the asymmetric induction (93:7 er), demonstrating the advantages of substrate recognition through attractive ionic interactions compared with steric biasing.

Substrates bearing 6′-alkenyl groups including cyclohexenyl (25) and 3,6-dihydro-2*H*-pyranyl (26), and 6′-chloro group (27–29) underwent the stereoselective reactions with various functionalized aryl boronic acids (up to 99.5:0.5 er). When 6′ position is unsubstituted, racemic product was isolated (30). We suspected that the reduced rotational barrier about N−C axis led to racemization at 60 °C, even if the product was formed stereoselectively. Indeed, appreciably enantioenriched product (91:9 er) was isolated when the reaction was performed at 40 °C for 1 h. The er eroded to 66.5:33.5 at 40 °C for 20 h, and the product racemized rapidly at 60 °C. Moreover, incorporating MeO groups adjacent to the reaction sites had an insignificant impact on enantioselectivity (31, 93:7 er). The results corroborate the notion that the catalyst does not rely on a sterically congested chiral pocket to exert stereoinduction.

Despite broad applications of N-aryl phenoxazines and N-aryl dihydroacridines in organic materials, their atropisomerism have remained unexplored. Reactions of phenoxazine derivative yielded 32

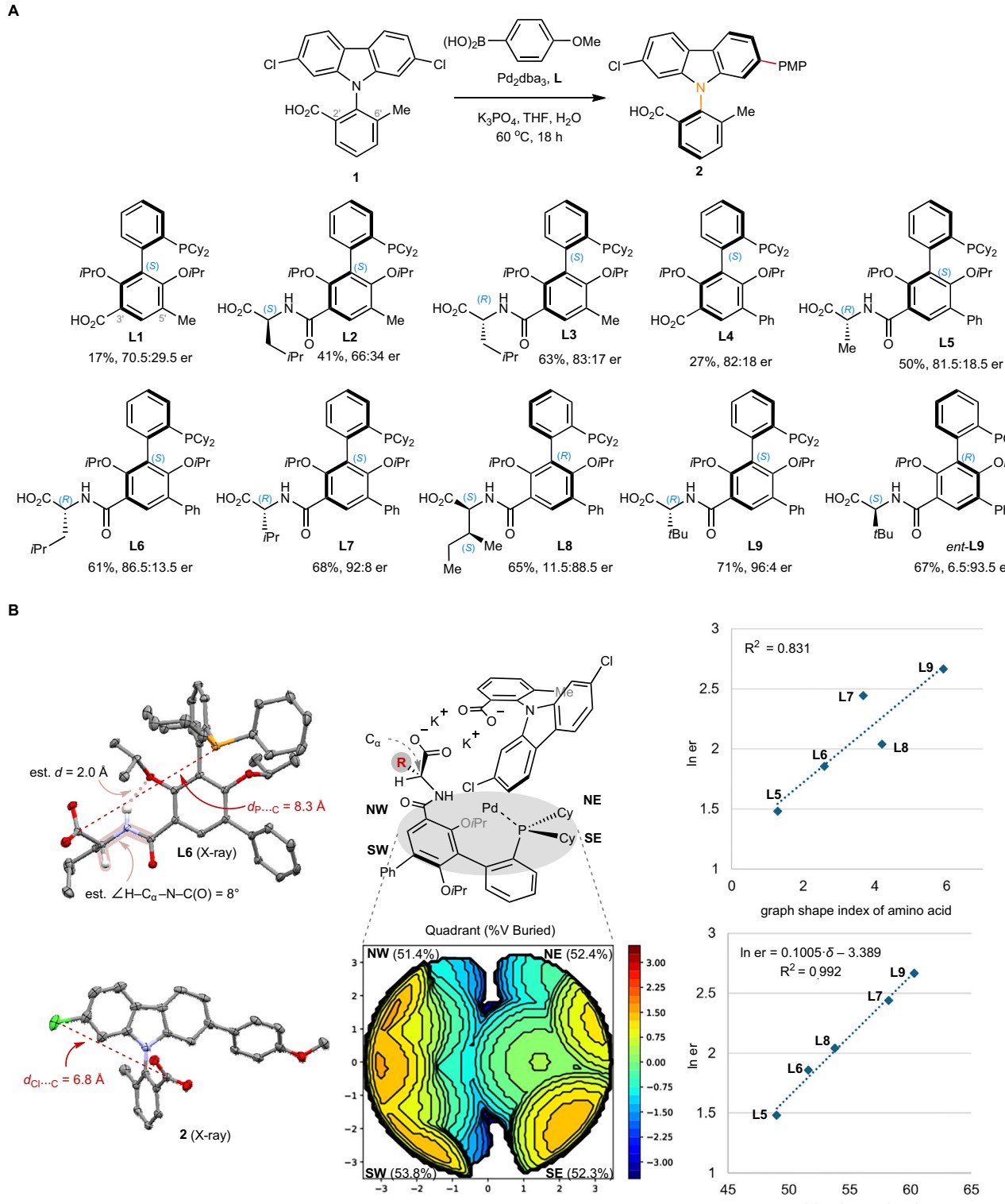

**Fig. 2 | Catalyst development and ligand parameterization. A** Survey of phosphine ligands. **B** Stereoinduction model based on absolute configurations, steric property map of **L6**, and parameterization analysis for ligands. Reaction conditions: **1** (0.1 mmol), PMP-B(OH)$_2$ (0.12 mmol), K$_3$PO$_4$ (0.5 mmol), Pd$_2$dba$_3$ (1.0 mmol%), Ligand (2.0 mmol%), THF (20 mL/mmol), H$_2$O (1.4 mL/mmol), 60 °C, 18 h. Enantiomeric ratio (er) was determined using HPLC–CSP. dba dibenzylideneacetone, Cy cyclohexyl, Me methyl, Ph phenyl, Pr propyl, PMP *p*-methoxyphenyl, THF tetrahydrofuran.

in 63.5:36.5 er at 60 °C and in 85.5:14.5 er at 50 °C. Enantiopurity of **32** likely eroded under the reaction conditions, given its ethyl ester racemized slowly at room temperature. The conformational flexibility of phenoxazine likely result in low rotational barriers about N−C axis.

By contrast, *N*-aryl 9,9-dimethyl-9,10-dihydroacridine **33** was synthesized in 91.5:8.5 er, demonstrating high conformational stability. In both cases, the chiral catalyst preserves effective stereocontrol despite changes in the substrate scaffolds.

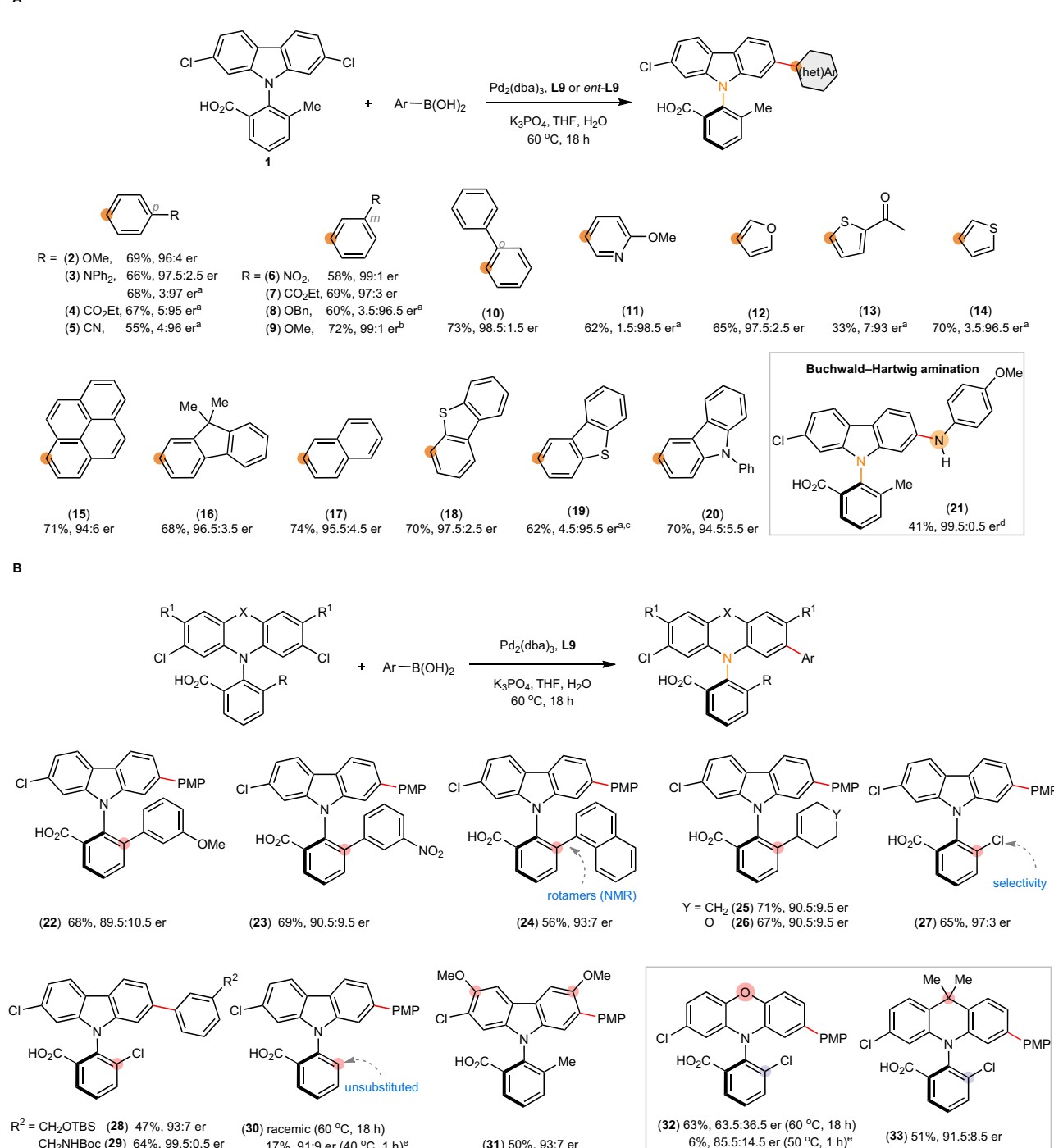

**Fig. 3 | Substrates scope of *N*-aryl carbazoles. A** Scope of coupling partners for Suzuki−Miyaura coupling and Buchwald−Hartwig amination. **B** Scope of *N*-aryl carbazole scaffolds. Reaction conditions: dichlorocarbazole (0.25 mmol), arylboronic acid (0.35 mmol), Pd₂(dba)₃ (1.0 mol%), **L9** (2.0 mol%), K₃PO₄ (1.25 mmol), THF (20 mL/mmol), H₂O (0.4 mL/mmol), 60 °C, 18 h. Isolated yield as ethyl ester reported. The absolute configurations of products were assigned by analogy to **2**. [a]With *ent*-**L9**. [b]1 mmol scale. [c]Isolated as methyl ester. [d]Buchwald−Hartwig amination was carried out with *p*-anisidine, K₃PO₄ (1.25 mmol), 1,4-dioxane (20 mL/mmol), 100 °C, 18 h. [e]With Pd₂(dba)₃ (2.0 mol %), **L9** (4.0 mol %), THF (2.5 mL/mmol). Bn benzyl; Boc *tert*-butyloxycarbonyl; Et ethyl; TBS *tert*-butyldimethylsilyl.

## Adaptability to N−N axial chirality

Aiming at a unified strategy, we probed the outcome of replacing N−C for N−N axes (Fig. 4A). *N*-pyrrol-1-yl-carbazole **34** coupled with a broad range of aryl boronic acids, affording the products in high stereoselectivity (**35**–**48**, 92:8−99.5:0.5 er). The absolute stereochemistry of **44** is consistent with that of *N*-aryl carbazole **2**, corroborating the determinant role of carboxylates. Interestingly, the N−N axis and the

carbazole moiety of **44** are not coplanar, contrasting with the coplanar N−C axis of **2**.

The chiral catalyst optimized for N−C substrate **1** was highly adaptable to N−N substrate **34**. Non-directional ionic interactions tolerate changes in the spatial arrangements of Cl and CO₂H. The results prompted us to test the catalyst's adaptability by shifting CO₂H to 3'-position of *N*-pyrrolyl moiety, further away from the N−N bond

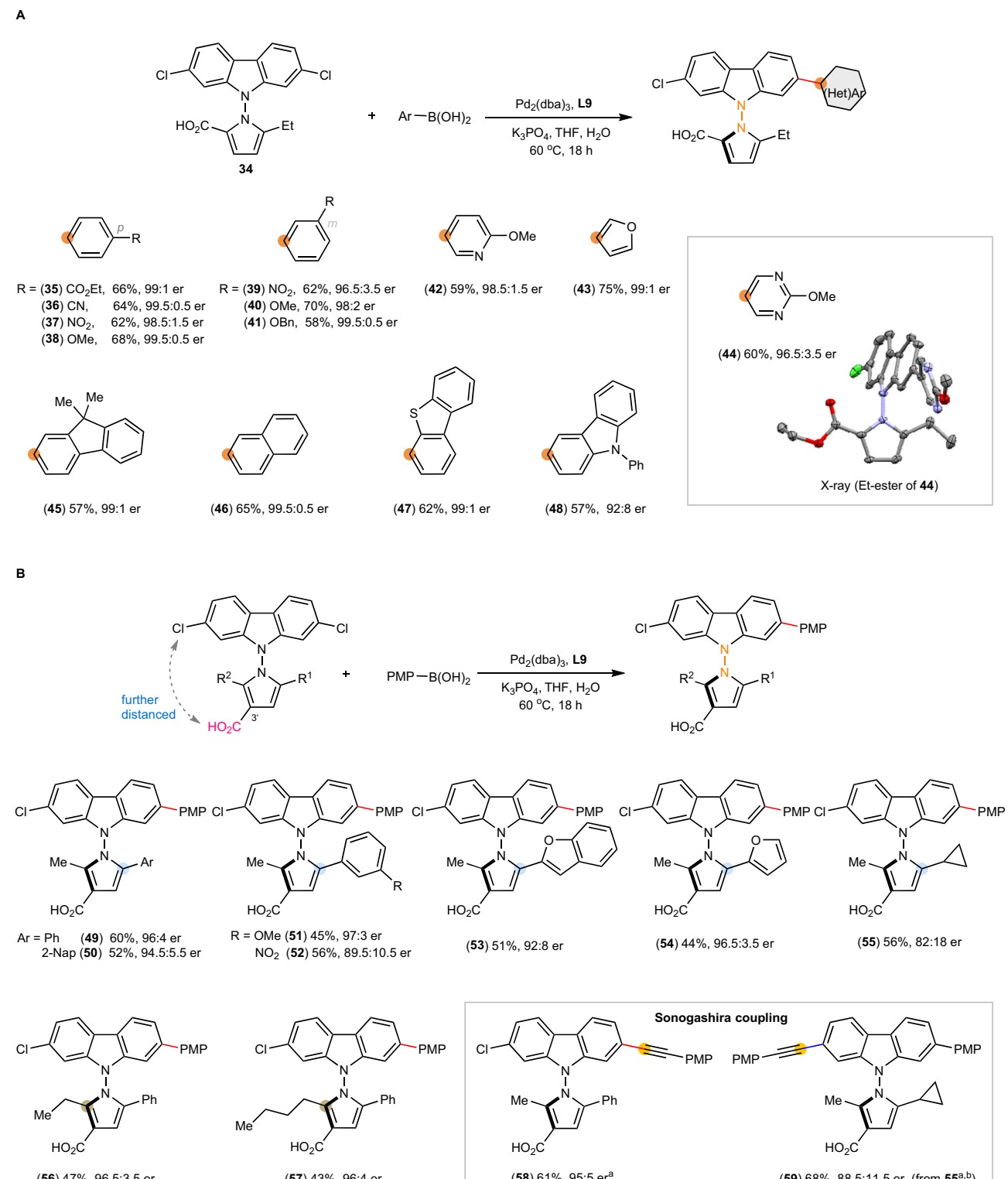

**Fig. 4 | Desymmetrization to access N–N stereogenic carbazoles. A** Aryl boronic acids. **B** The adaptability to further distancing of carboxyl group. Reaction conditions: same as Fig. 3. The absolute configuration of **44** was assigned by X-ray crystallography. The absolute configurations of other products were assigned by analogy. [a]Sonogashira coupling was carried out with 4-ethynylanisole. [b]With *ent*-**L9**. Nap: naphthyl.

(Fig. 4B). The same catalyst remained effective in spite of the pronounced structural change (**49**, 96:4 er). The 3'-$CO_2H$ is stereo-determining: masking it as an ethyl ester led to formation of racemic **49** (51:49 er). The results also showed that simple steric biasing at *ortho* positions of N–N bond (Ph vs Me) is insufficient in directing the remote stereocontrol.

Diverse products (**50–54**) were obtained in 89.5:10.5–97:3 er using substrates bearing various (hetero)aryl groups at the 5-position of the pyrrolyl moiety. While modest er was obtained with a 5-cyclopropyl substituent (**55**, 82:18 er), 2-Et and 2-Bu did not impact the enantioselectivity (**56**, 96.5:3.5 er; **57**, 96:4 er). Furthermore, Sonogashira coupling proceeded in 95:5 er (**58**). When **55** (82:18 er) was subjected to

Sonogashira coupling using enantiomer of **L9**, bisfunctionalized product **59** was isolated in an improved 88.5:11.5 er. Plausibly, a kinetic resolution of **55** was operating.

## Synthetic utilities

The new strategy enables modular access to functionalized N–C and N–N axially chiral carbazoles through sequential coupling (Fig. 5). To illustrate, 3,6-dichloro N-pyrrolyl carbazole **60** underwent Suzuki–Miyaura reaction in 83.5:16.5 er (**61**, Fig. 5A). This class of N–N axially chiral carbazoles exhibits excellent configurational stability. No erosion in enantiopurity was observed even after refluxing **42** and **61** in toluene for 24 hours. Subsequent reaction of **61** using the enantiomer of catalyst supported by (*R*)-**L4** yielded product **62** in 93.5:6.5 er. The improvement in enantiopurity is attributed to a secondary kinetic resolution, which favors the major enantiomer of **61** during the second coupling reaction by inducing a configuration opposite to that of the initial step. Electronically differentiated arenes were incorporated at remote sites, and the stereochemistry of N–N axis was established concurrently. In contrast, 3,6-dichloro N-aryl carbazole underwent the desymmetrization reaction in only 69.5:30.5 er (see SI, SI-**10**).

Enantio- and site-selective coupling of trichloro-substituted **63** afforded **64** in 97.5:2.5 er (Fig. 5B). Subsequent coupling with excess aryl boronic acid using SPhos as ligand yielded **65** without change in the enantiopurity (97.5:2.5 er). However, complex product mixture was obtained when stoichiometric quantity of aryl boronic acid was used, presumably due to low selectivity between two remaining chloro groups of **64**. This issue was overcome by employing the enantiomeric chiral ligand, which led to formation of **66** in 77% yield. The heterofunctionalized carbazole **67** (97.5:2.5 er) was successfully synthesized upon a final coupling. The sequential coupling approach provides an efficient synthetic method for enantioenriched monomers and tactic oligomers. AB-type monomer **71** (95:5 er) was prepared by coupling **68** to 4-B(dan) phenylboronic pinacol ester **69**, and subsequent Pd-catalyzed polycondensation yielded tactic conjugated oligomer **72** (Fig. 5C).

The carboxyl group serves as a versatile handle for chemical derivatization. The coupling reaction using amine-substituted boronic acid **73** afforded **74** in 99:1 er. Upon *N*-deprotection, the resulting amine–carboxylic acid intermediate was subjected to amide coupling conditions. Because of the rigid *N*-aryl carbazole backbone, intramolecular lactamization was not observed, and a 24-membered macrocyclic dimer **75** featuring two N–C chiral axes was obtained with >99.5:0.5 er (Fig. 5D). Curtius rearrangement of acyl azide derived from **9** in *t*BuOH yielded Boc-protected amine **76** in high enantiopurity (99:1 er), along with a novel $C_2$-symmetric chiral urea **77** (Fig. 5E). The DIBAL-H reduction of **3** and *ent*-**3** to benzyl alcohols followed by subsequent oxidative condensation with benzimidamide afforded 1,3,5-triazine-substituted *N*-aryl carbazoles **78** (98:2 er) and *ent*-**78** (3:97 er), respectively. Upon Pd-catalyzed proto-dehalogenation of **78**, the deschloro analog **79** (98.5:1.5 er) was successfully prepared (Fig. 5F).

Finally, desymmetrization reaction of 2,7-dibromocarbazole (**80**) proceeded in 92.5:7.5 er (**81**), affording 2,7-disubstituted *N*-aryl carbazoles **83** in 92:8 er after a subsequent cross-coupling with **82** (Fig. 5G). The results further demonstrates the applicability of the desymmetrization strategy.

## Chiroptical properties

The modular synthesis of enantioenriched carbazoles provided an opportunity to evaluate carbazole atropisomers as potential chiral chromophores. We first studied their photophysical properties in diluted solution (see SI, Fig. S1–S3). They exhibit violet to blue emission ($\lambda_{PL}$ = 354–425 nm) with nanosecond-scale lifetime. The N–N compounds emit at shorter wavelengths than the corresponding N–C compounds, introducing a distinct dimension to tuning fluorescence beyond modifications of the peripheral substituents. The absolute

quantum yields range from moderate to high ($\Phi$ up to 82.4% for *S*-**5**), demonstrating their excellent photophysical properties as models for chiroptical functions.

We next investigated the ground-state chiroptical properties in solution. The circular dichroism (CD) spectra exhibits active Cotton effects (see SI, Fig. S4). The dissymmetry factors ($g_{abs}$ up to 5.36×10$^{-4}$ for *S*-**53**) indicate a moderate degree of circular polarization, which is expected for typical chiral organic molecules.

Furthermore, the Cotton effects are consistent between analogous N–C and N–N stereogenic carbazoles (i.e., positive Cotton effects observed for *R*-isomers at $\lambda_{max}$, Fig. 6A and Table S2). The calculated CD spectra of *S*-**4** and *S*-**5** optimized using density functional theory (DFT) shows negative Cotton effects at the maximum absorbances, which aligns well with the experimental data (Fig. 6B–D and see SI, Fig. S5).

Chiral fluorescent molecules' unique ability to emit left and right circularly polarized lights with different intensities have garnered increasing attention as photonic and electrooptical materials[59,60]. The luminescence and CD results motivated us to investigate the photo-excited chiral properties in thin-film state. By dispersing *S*-**5** in polymethyl methacrylate (PMMA), we observed an *r*-CPL signal at the highest emission wavelength (410 nm) with a $g_{lum}$ value of −0.004 (Fig. 6E). Importantly, compounds including N–C carbazole **4**, N–C acridine **33**, as well as N–N carbazoles with π-conjugated electron-deficient arene (**35**), electron-rich arene (**38**), and heteroarene (**44**) all exhibit CPL activities (see SI, Fig. S6 and Fig. S7). Scaffold-tuned maximum emission wavelengths were observed for N–N carbazoles (ca. 380 nm), N–C carbazoles (ca. 410 nm), and N–C acridine (518 nm).

Besides structural modifications, the electron-rich carbazole offers opportunity in achieving charge transfer-modulated chiroptical functions. To investigate this noncovalent strategy, we introduced tetracyanobenzene (TCNB) as an electron-acceptor. Fabricating *S*-**5** and TCNB (1:1 molar ratio) in PMMA afforded orange luminescence (580 nm), and the CPL property was preserved with a similar $g_{lum}$ value (Fig. 6E).

DFT optimization revealed the exquisite π–π stacking between carbazole of *S*-**5** and TCNB ($d$ = 3.44 Å, ΔE = -15.64 kcal/mol, Fig. 6F). The computed HOMO–LUMO energy gap narrows upon formation of the charge-transfer complex (2.46 eV vs. 4.06 eV), leading to bathochromic-shifted emissions for CT-CPL (Fig. 6G). The CT-CPL strategy is applicable to N–C carbazole **4**, as well as N–N carbazoles coupled with π-conjugated electron-deficient arene (**35**) and electron-rich arene (**38**) (see SI, Fig. S8–S10).

Molecules that possess both TADF and CPL activities are particularly appealing for achieving highly efficient circularly polarized luminescence[61]. The potential of axially chiral carbazoles as functional luminescent materials was further illustrated by the CP-TADF activity of **78** and **79** (Fig. 6H). The through-space electron transfer between triazine and carbazole moieties could facilitate the intersystem crossing (ISC)[62]. Using bis[2-(diphenylphosphino)phenyl] ether oxide (DPEPO) as the host material, the emission curve of **78** exhibited a prolonged lifetime of 6.34 μs (Fig. 6I). The prompt and delayed emission spectra display identical shape and peak locations (Fig. 6J), indicating delayed fluorescence rather than phosphorescence[63]. The TADF activity was further evidenced by temperature-variable emission spectra from 80 K to 380 K (Fig. 6K). As the temperature increased from 80 K to 220 K, the emission of **78** increases, contrasting normal luminophores (Fig. 6L). The reversal ISC (RISC, T$_1$→S$_1$) could be facilitated at elevated temperature, thereby improving the efficiency of TADF. Increasing the temperature further led to nonradiative transitions through skeleton motions, resulting in reduced RISC. The deschloro analogs **79** also exhibited TADF phenomenon with similar luminescence properties (see SI, Fig. S11). DFT calculation of the excited states of **78** showed a narrow gap between S$_1$ and T$_1$ (ΔE$_{st}$ = 0.23 eV), leading to accelerated ISC and RISC processes. Furthermore,

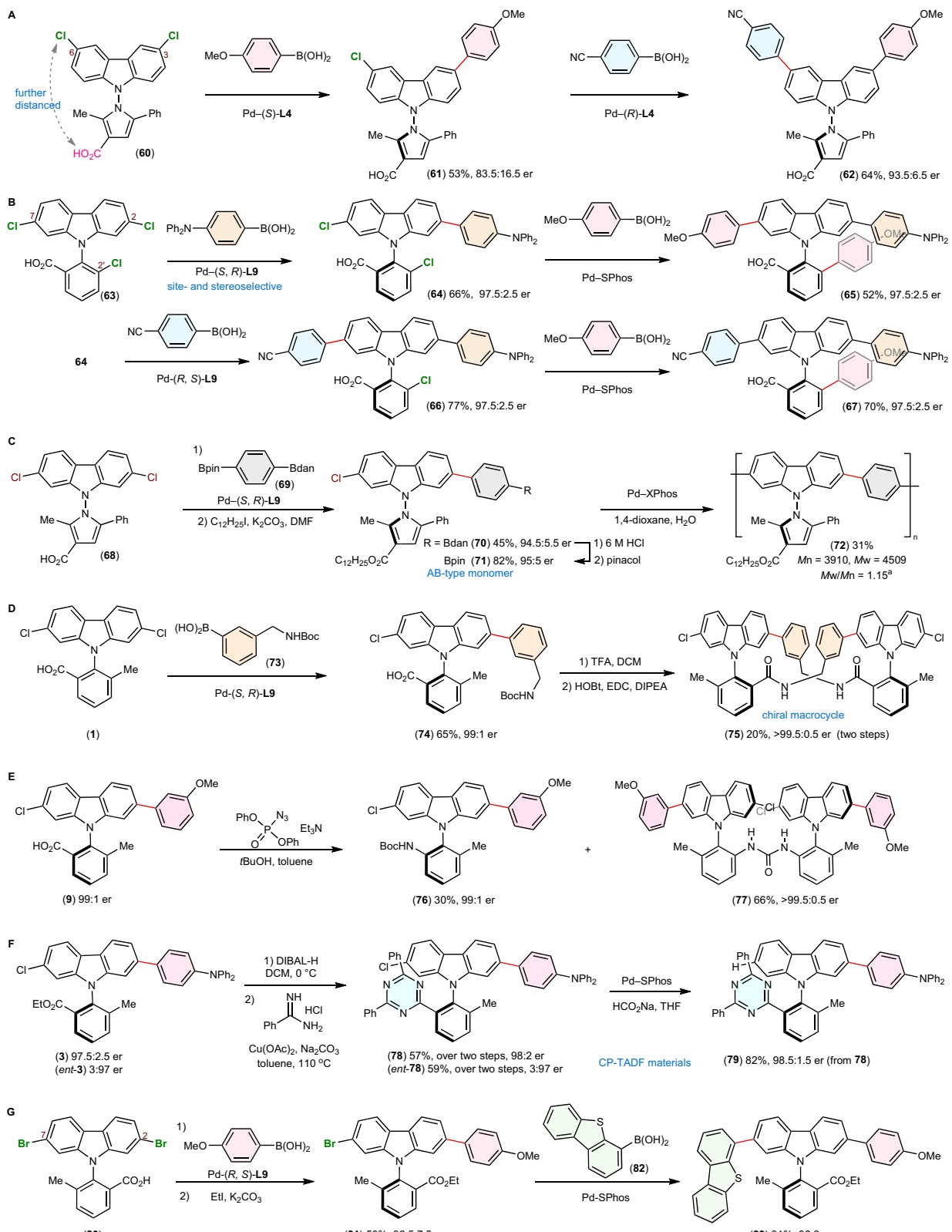

**Fig. 5 | Modular access to functionalized carbazole atropisomers. A** Synthesis of 3,6-disubstituted carbazoles. **B** Sequential coupling of trichloro *N*-aryl-carbazole. **C** Tactic conjugated oligomer. **D** 24-Membered chiral macrocycle. **E** Curtius rearrangement to amino compounds. **F** Access to new CP-TADF materials. **G** Synthesis of 2,7-disubstituted carbazoles through desymmetrization of 2,7-dibromocarbazole **80**. [a]*M*n, *M*w, and PDI were determined by gel permeation chromatography

(GPC). Ac: acetyl; dan: naphthalene-1,8-diaminato; DIBAL-H: diisobutylaluminum hydride; DIPEA: *N*,*N*-diisopropylethylamine; EDC: *N*-(3-dimethylaminopropyl)-*N*′-ethylcarbodiimide; HOBt: 1-hydroxybenzotriazole; pin: pinacolato; SPhos: 2-dicyclohexylphosphino-2′,6′-dimethoxybiphenyl; TFA: trifluoroacetic acid; XPhos: 2-dicyclohexylphosphino-2′,4′,6′-triisopropylbiphenyl.

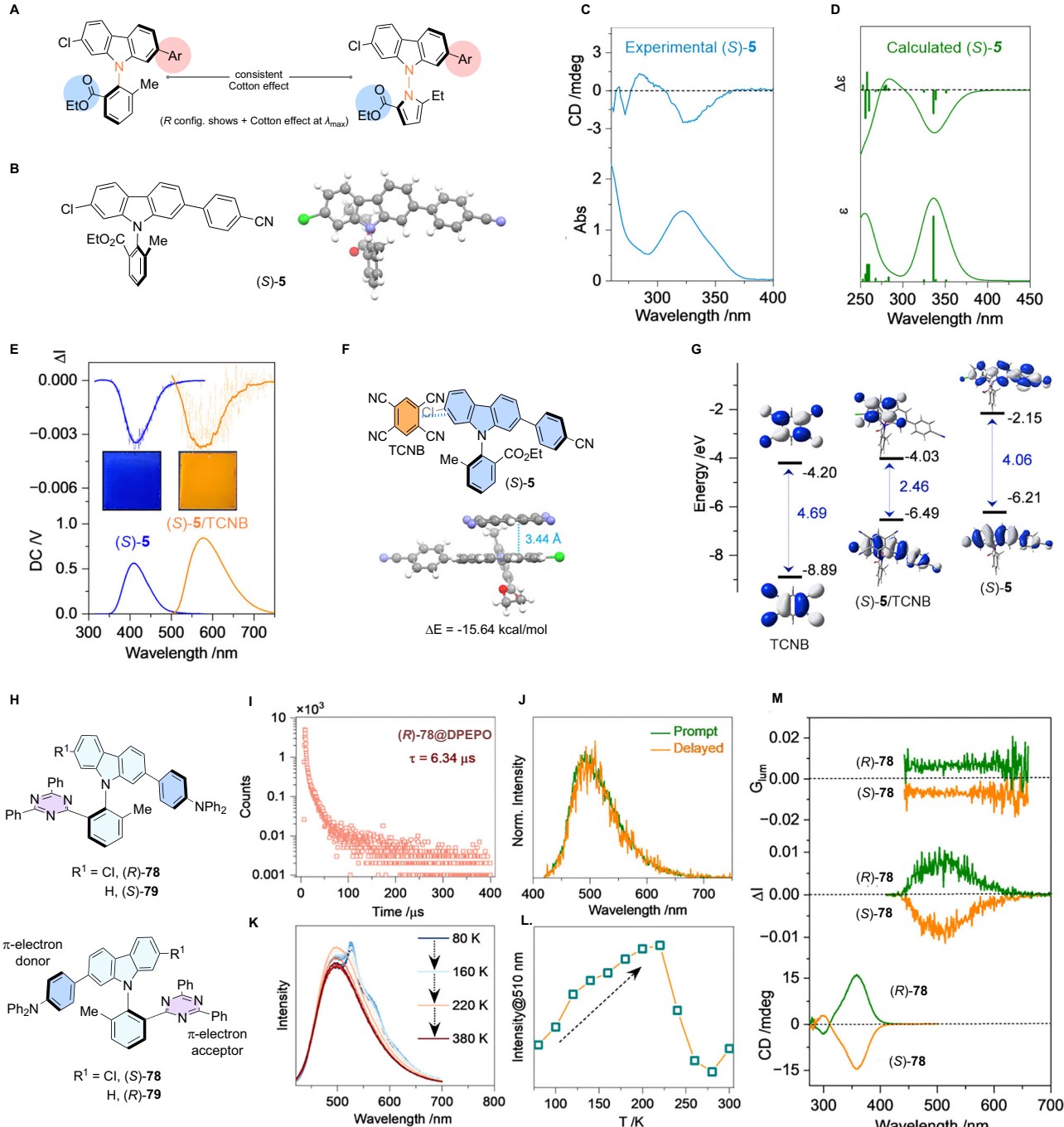

**Fig. 6 | Circular dichroism (CD), circularly polarized luminescence (CPL), and circularly polarized thermally activated delayed fluorescence (CP-TADF) of axially chiral carbazoles. A** Cotton effects are consistent between analogous N−C and N−N atropisomers. **B** DFT optimized structure of (S)-**5** for CD calculation. **C** CD spectrum of (S)-**5** in 1,2-dichloroethane (c = 0.2 mM). **D** Calculated electronic CD spectrum of (S)-**5**. **E** CPL spectra of (S)-**5** without and with benzene-1,2,4,5-tetra-carbonitrile (TCNB) in polymethyl methacrylate (PMMA) matrices ($\lambda_{ex}$ = 300 nm and 440 nm, respectively). Inserts: emissive photographs. **F** DFT-optimized complexation of (S)-**5** and TCNB. **G** Frontier orbital of individual species and complex.

**H 78** and **79** for CP-TADF studies. **I** Delayed emission curve of (R)-**78** in bis[2-(diphenylphosphino)phenyl] ether oxide (DPEPO) matrix. **J** Prompt and delayed emission spectra of (R)-**78** ($\lambda_{ex}$ = 400 nm, delayed time = 0.1 ms). **K** Temperature-variable emission spectra of (R)-**78** in DPEPO matrix. **L** Temperature-dependency of emission ($\lambda$ = 510 nm). **M** CD, CPL and $g_{lum}$ curves of (R)-**78** (CD was measured in 1,2-dichloroethane, c = 0.2 mM, CPL was measured in DPEPO matrix. Also see Fig. S1–S10 for additional CD and CPL data of N−C and N−N carbazoles, and Fig. S11 and Fig. S12 for the of CP-TADF data of (R)−**79** and (S)−**79**.

strong Cotton effects were observed for **78** in solution, reflecting conformational rigidity of its molecular scaffold (Fig. 6M). When doped in DPEPO, R-**78** and S-**78** exhibited fine mirrored CPL signals, giving rise to l- and r-CPL respectively (500 nm, $g_{lum} \pm 0.008$). Meanwhile, S-**79** and R-**79** exhibited l- and r-CPL respectively with comparable $g_{lum}$ values (see SI, Fig. S11 and Fig. S12). The chiroptical

handedness l- and r- is determined by the relative stereochemistry of the luminophores rather than the R/S configurations.

## Discussion

In summary, we have developed a unified enantiotopic-group-selective strategy to access highly enantioenriched carbazole atropisomers and

established their functions as chiral chromophores. We addressed the synthetic challenge of remote atroposelectivity through leveraging the fundamental phenomena of ionic interactions. The ionic catalyst system maintained effective stereocontrol while enabling the incorporation of diverse functionalities into a variety of N–C and N–N axially chiral scaffolds. The high substrate adaptability allowed for tailored synthesis of carbazole-, phenoxazine-, and dihydroacridine-derived atropisomers that are inaccessible through conventional methods. We found that establishing stereogenicity around the nitrogen atom of the carbazole core is an effective approach for conferring chiroptical functions. Cotton effects are consistent between N–C and N–N analogs, with the latter exhibiting blueshifted emissions. CPL activities could be modulated (380 nm to 580 nm) through structural diversification (e.g., CP-TADF) and noncovalent complexation (e.g., CT-CPL). Given the widespread applications of π-systems in organic materials and the importance of cross-coupling reactions in their functionalization, we anticipate that catalyst engineering will facilitate long-range stereoinduction in extended π-systems for function-oriented applications. Future studies on nitrogen-centered chiral chromophores will stimulate their applications in photonic and optoelectronic materials for future computing, sensing, and display technologies.

## Methods

### General

**procedure for desymmetrizing Suzuki–Miyaura cross-coupling reaction.** Under $N_2$ atmosphere, $Pd_2(dba)_3$ (1.0 mol%) and **L9** (2.0 mol%) were dissolved in THF (0.5 mL), and the resulting mixture was then stirred at room temperature for 20 min. The resulting solution was added to a reaction flask containing dichlorocarbazole **1** (0.25 mmol), arylboronic acid (0.35 mmol), $K_3PO_4$ (1.25 mmol), THF (4.5 mL) and $H_2O$ (0.10 mL). The reaction mixture was then sealed under $N_2$ atmosphere and heated to 60 °C while stirred using a magnetic stirring bar for 18 h. Upon reaction completion, the mixture was neutralized to pH 3–5 with 1.0 N aqueous solution of HCl and then extracted with ethyl acetate (4.0 mL) three times. The combined organic phase was washed with brine, dried over $Na_2SO_4$, and concentrated in vacuo to give the crude product. The desired cross-coupling product was purified using flash column chromatography on silica gel. In cases that ester products were isolated, a Schlenk-tube was charged with the crude product of the cross-coupling reaction, followed by addition of *N,N*-Dimethylformamide (1.0 mL), EtI or MeI (0.5 mmol) and $K_2CO_3$ (1.0 mmol). The reaction mixture was then stirred overnight at room temperature. The resulting mixture was diluted with ethyl acetate (20 mL) and washed with water, brine, and dried over $Na_2SO_4$. The organic layer was concentrated in vacuo, and the desired product was purified using flash column chromatography on silica gel.

## Data availability

The authors declare that all data generated and analyzed in this study are available within the article and its Supplementary Information file. For experimental details, spectra for all new compounds, and chiroptical properties data associated with all tables and figures, please see Supplementary Information. The X-ray crystallographic coordinates for structures reported in this study have been deposited at the Cambridge Crystallographic Data Center (CCDC) under deposition number 2390746 (**2**), 2390747 (ethyl ester of **44**) and 2390748 (**L6**). These data can be obtained free of charge from The Cambridge Crystallographic Data Center via (www.ccdc.cam.ac.uk/data_request/cif). All data are available from the corresponding authors upon request.

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

## Acknowledgements

We thank Geok Kheng Tan (NUS) and Irwan Iskandar Bin Roslan (NUS) for solving X-ray crystal structures. This research was supported by Singapore Ministry of Education (Academic research fund MOE-T2EP10222-

0004) (Y.Z.). We also thank the National Natural Science Foundation of China for financial support (No. 22171165, 22371170) (P.X.).

## Author contributions

J.W. and Z.W. performed the experiments. Y.Z. and P.X. directed the project and wrote the first draft of the manuscript. All discussed the results and commented on the manuscript. J.W. and Z.W. contributed equally to the work.

## Competing interests

The Authors declare no competing interests.
