## [Transparent Peer Review file · Nature Communications]

Enantiotopic-Group-Selective Coupling for Unified Access to Carbazole Atropisomers as Versatile Chiral Chromophores

Corresponding Author: Professor Ye Zhu

Version 0:

Reviewer comments:

Reviewer #1

(Remarks to the Author)

Xing, Zhu, and coworkers developed the selective synthesis of enantioenriched carbazole atropisomers. They performed the Suzuki-Miyaura coupling of N-aryl-dichlorocarbazoles with chiral ligands to give enantioenriched carbazole atropisomers. Although the authors performed many experiments and showed wide substrate scope, present reactions do not have generality; in other words, most of other researchers will not use the reaction, partly because the dichlorocarbazoles and chiral ligands are difficult to prepare. In addition, there are many reports on atroposelective synthesis and some examples for the synthesis of enantioenriched indole/carbazole atropisomers. Only the modest chiroptical properties were shown probably because the compounds are not well designed for the chiroptical materials, which is not suitable to the high impact journals. Therefore, the reviewer recommend the publication of this manuscript in more specialized journal after the several modifications.

The authors are advised to include the results of the corresponding dibromocarbazole(s).

It may be interesting to investigate the chiroptical properties of polymer 72 and to compare the results with those of the monomer(s).

SI

Are the following abbreviations correct?

Bu buty

bpin boronic acid pinacol

TBS tert-butyloxycarbonyl

Reviewer #2

(Remarks to the Author)

Reviewer #3

(Remarks to the Author)

Zhu and Xing present an atroposelective cross-coupling of carbazoles to access C–N axially chiral biaryl compounds, with the methodology further extended to N–N axially chiral systems. The reaction proceeds predominantly at the C-2 position of carbazole, with one example demonstrating functionalization at C-3 in the synthesis of N–N axially chiral compound. The scope of the transformation is explored, and the resulting products are derivatized into polyarenes, with their chiroptical properties analyzed. While the work is suitable for publication, the following revisions are recommended:

1. It is suggested to revise the abstract to more accurately reflect prior literature, particularly regarding the synthesis of C–N axially chiral carbazoles. For instance, it is claimed that "chiroptical applications of their atropisomers have remained untapped due to the synthetic challenge of achieving remote atroposelectivity" and that the work "addresses the long-

standing synthetic challenge of remote atroposelectivity" should be polished, given that a catalytic method for atroposelective C–N bond formation on carbazoles have been previously reported (ref. 13). A more balanced discussion, acknowledging prior contributions while explaining the new advances in this work, is needed.

2. Given the structural similarities between the chiral carbazoles reported here and those in prior literature, a more detailed discussion (supported by a graphic in Figure 1) should be included to clearly delineate the new aspects of this study. The authors should explicitly compare their findings with the precedent case (ref. 13), explaining the key differences in methodology.

3. While the coupling reaction is primarily demonstrated at C-2 of carbazole, one example shows C-3 coupling to give N-N axially chiral compound. Can the reaction be applied to C-3 functionalization to form C–N axial chirality?

4. In the transformation from 61 to 62, the e.r. increases. Is this step a kinetic resolution? A proper discussion is needed. Since 61 is already chiral, an achiral ligand might suffice for its conversion to 62. The authors should explain why a chiral ligand is necessary in this step and provide additional commentary in the main text to clarify the mechanistic rationale.

Version 1:

Reviewer comments:

Reviewer #2

(Remarks to the Author)

Reviewer #3

(Remarks to the Author)

In this revised manuscript, the authors fully addressed the concerns raised by the reviewer and publication is recommended.

Reviewer #1 (Remarks to the Author)

Overall Comments: Xing, Zhu, and coworkers developed the selective synthesis of enantioenriched carbazole atropisomers. They performed the Suzuki-Miyaura coupling of *N*-aryl-dichlorocarbazoles with chiral ligands to give enantioenriched carbazole atropisomers. Although the authors performed many experiments and showed wide substrate scope, present reactions do not have generality; in other words, most of other researchers will not use the reaction, partly because the dichlorocarbazoles and chiral ligands are difficult to prepare. In addition, there are many reports on atroposelective synthesis and some examples for the synthesis of enantioenriched indole/carbazole atropisomers. Only the modest chiroptical properties were shown probably because the compounds are not well designed for the chiroptical materials, which is not suitable to the high impact journals. Therefore, the reviewer recommends the publication of this manuscript in more specialized journal after the several modifications.

Our Response:

We sincerely thank the Reviewer for their careful evaluation of our work and for the constructive comments provided. We would like to address each point as follows:

(1) The Reviewer made an excellent point regarding the importance of the applicability of synthetic methods, particularly the accessibility of substrates and catalysts, which can determine whether other researchers adopt these methods. Indeed, although a large number of chemical transformations have been developed, only a few are widely utilized. Among these, Pd-catalyzed cross-coupling reactions are the most commonly used reactions in conjugated materials and are second only to amide synthesis in medicinal chemistry. Therefore, we chose to engineer chiral catalysts for cross-coupling reactions, with the aim of increasing the likelihood of adoption, as this approach does not require changes to existing synthetic routes.

Although the carbazole substrates are not commercially available, they can be prepared in 2–4 steps from commercially available starting materials through nucleophilic aromatic

substitution (56% yield for model substrate, SI, page S13–S14). We acknowledge that the ligand synthesis is somewhat lengthier, and we can prepare the ligands on gram scale in the laboratory. The starting RuPhos is readily available and affordable (<\$600 for 0.5 kg), as well as the amino acids. We are actively working to streamline the ligand synthesis for scaling-up via chiral resolution. Recently, we have collaborated with a pharmaceutical company to demonstrate the applicability of our ligands in drug molecule synthesis, which also drives us to produce the ligands in larger quantity. We are optimistic that our effort should further facilitate their application.

Again, we greatly appreciate the Reviewer's incisive comment regarding material availability, which resonates with my own experience as a process chemist in the pharmaceutical industry.

(2) We thank the Reviewer for their comments regarding the prevalence of atroposelective reactions. Indeed, atroposelective synthesis has been an area of active research. However, previous studies have primarily focused on indole atropisomers, with only a few reports on carbazole-based atropisomers (references 11, 12, 40). In particular, the generation of stereogenicity through differentiated substituents at the remote 2,7-positions on carbazoles has not been reported to the best of our knowledge. Furthermore, our work represents the first example of long-range enantioselective desymmetrization of N-aryl carbazoles, enabling access to both C–N and N–N axially chiral frameworks. We believe that our work provides a new approach to the synthesis of carbazole atropisomers, which are far less available than their indole analogues.

(3) We appreciate the Reviewer's forward-looking suggestion that the chiroptical properties could be further optimized through structural design. In the current manuscript, our primary focus has been on the development of the synthetic methodology. We have used the molecules prepared during our substrate scope exploration in methodology development to illustrate the diverse chiroptical activity including CD, CPL, CT-CPL, CP-TADF, and CPL-TADF. We hope that our study provides a proof-of-concept that C–N and N–N axially chiral carbazole scaffolds can serve as versatile platforms for further materials design.

Again, we thank the Reviewer for encouraging us to explore further opportunities to enhance molecular properties through rational design.

Comment 1:

The authors are advised to include the results of the corresponding dibromocarbazole(s).

Our Response:

Thank you for the incisive advice that led to expansion of the scope of our study during the revision. We have tested the corresponding dibromo-carbazole substrate under our optimized reaction conditions. The desired mono-coupled product (**81**) was obtained in high er (compound **81** 92.5:7.5 er) with *p*-methoxyphenylboronic acid. We have also successfully coupled the remaining bromo group for dibenzothiophene (compound **83**) to illustrate the application of this approach. We have revised the manuscript accordingly (see Fig. 5G, pages 11 and 12; SI, pages S98–S101).

On the third paragraph of page 12.

“Finally, desymmetrization reaction of 2,7-dibromocarbazole (**80**) proceeded in 92.5:7.5 er, affording 2,7-disubstituted N-aryl carbazoles **83** in 92:8 er after a subsequent cross-coupling (Fig. 5G). The results further demonstrates the applicability of the desymmetrization strategy.”

Comment 2:

It may be interesting to investigate the chiroptical properties of polymer **72** and to compare the results with those of the monomer(s).

Our Response:

Thank you very much for the insightful suggestions. As part of an ongoing collaboration, we are investigating the chiroptical properties of oligomer **72** and its analogues' anions following hydrolysis. We are utilizing these compounds as conjugated oligoelectrolytes to study their

fluorescence properties and chiral optical applications. We look forward to reporting the results of this research in the future.

Comment 3:

SI

Are the following abbreviations correct?

Bu buty; bpin boronic acid pinacol; TBS *tert*-butyloxycarbonyl

Our Response:

Thank you very much for advising us. We sincerely apologize for our mistakes and have corrected them accordingly. We have also thoroughly examined other parts of the SI to correct mistakes.

On pages S3 and S4 in SI.

“Bu Butyl; Bpin Pinacol boronic ester; TBS *tert*-Butyldimethylsilyl”

Reviewer #2 (Remarks to the Author)

Overall Comments: *N*-aryl carbazoles have wide applications in visible-luminescent materials, the chiroptical applications of this kind of organic compounds were seldom reported. In this manuscript, Xing and Zhu described an efficient Pd-catalyzed asymmetric cross-coupling desymmetrization to construct *N*-aryl carbazoles and *N*-pyrrole carbazoles containing C-N and N-N axially chirality with excellent enantioselectivities, and further applied them to circular dichroism(CD), circularly polarized luminescence (CPL), charge-transfer CPL (CT-CPL) and circularly polarized thermally activated delayed fluorescence (CP-TADF), which provide a good solution for classical organic synthesis. The results are valuable and of great interest to readers. The manuscript is well written and the substrate scope has been well investigated. In my opinion, this study is valuable for the publication in *Nat. Commun.* after some minor revisions are carried out.

Our Response:

We thank the Reviewer for their constructive comments and support for the publication of this work.

Comment 1:

How about 1,8-dichloro substituted *N*-Aryl carbazole substrate carboxyl group, for example 1,8-dichloro-9-(3-methyl-[1,1'-biphenyl]-2-yl)-9*H*- carbazole.

Our Response:

We appreciate this insightful suggestion. Unfortunately, after trying several methods, we were unable to prepare the 1,8-dichloro-substituted *N*-Aryl carbazole substrates successfully. Intriguingly, this type of 1,8-dichloro-substituted *N*-Aryl carbazoles has not been reported previously.

To prevent chlorination at 3- and 6-positions, methyl groups were introduced to block these sites to achieve 1,8-chlorination. However, due to significant steric hindrance, under basic conditions the 1,8-dichloro carbazole (compound **a**) failed to undergo the S_NAr reaction efficiently.

Subsequently, we modified our strategy. The S_NAr reaction using unchlorinated carbazole proceeded smoothly, affording compound (d) in high yield. However, during the chlorination step under several different conditions including acid media and nucleophilic catalysis, the reactions were very complex, and we were unable to synthesize the 1,8-dichloro-substituted *N*-Aryl carbazoles. In one case (NCS in TFA/DCM), we obtained the mono-chlorination product in <10% yield out of a complex reaction mixture; however, we were unable to further chlorinate this compound without decomposition.

Comment 2:

Some remarkable works relate to the constructing of the C-N axially chiral compounds should be referenced.

Our Response:

Thank you for advising us. We have included several leading references on C-N axially chiral compounds as follows:

On pages 18 and 19 in the references.

“25. Gu, X.-W. et al. Stereospecific Si-C coupling and remote control of axial chirality by enantioselective palladium-catalyzed hydrosilylation of maleimides. *Nat. Commun.* **11**, 2904 (2020).”

“28. Li, Y., Liou, Y.-C., Oliveira, J. C. A. & Ackermann, L. Ruthenium(II)/imidazolidine carboxylic acid-catalyzed C–H alkylation for central and axial double enantio-induction. *Angew. Chem. Int. Ed.* **61**, e202212595(2022).”

“29. Dai, L. et al. Diastereo- and atroposelective synthesis of *N*-arylpyrroles enabled by light-induced phosphoric acid catalysis. *Nat. Commun.* **14**, 4813 (2023).”

“32. Ren, P., Zhao, Q., Xu, K. & Zhu, T. Enantioselective *N*-heterocyclic carbene-catalyzed Hauser-Kraus annulations for the construction of C–N axially chiral phthalimide derivatives. *ACS Catal.* **14**, 13195–13201 (2024).”

Please let us know if there are particular examples that we might have overlooked. Thank you again.

Comment 3:

How about the racemization energy barrier of the chiral compound **61**? Suggest the author to check the half-life.

Our Response:

Thank you for the suggestions to evaluate the racemization of this new class of molecules. We have examined the configurational stability of products **11**, **42** and **61** under rigorous conditions (toluene, 120 °C, 24 h). In both cases, no racemization was observed, confirming that the compounds are highly stable.

Each enantioenriched sample was dissolved in toluene and then heated up at the specified temperature (80, 100, and 120 °C) for 24 hours. No racemization was observed in all cases.

Therefore, the erosion of enantiopurity is unlikely under the typical reaction conditions (60 °C).

We have included a sentence in the main text of the revised manuscript (see below) and included the results in the revised SI, page S77.

On the third paragraph of page 10.

“This class of N–N axially chiral carbazoles exhibits excellent configurational stability. No erosion in enantiopurity was observed even after refluxing **42** and **61** in toluene for 24 hours (see SI for details).”

Reviewer #3 (Remarks to the Author):

Overall Comments: Zhu and Xing present an atroposelective cross-coupling of carbazoles to access C–N axially chiral biaryl compounds, with the methodology further extended to N–N axially chiral systems. The reaction proceeds predominantly at the C-2 position of carbazole, with one example demonstrating functionalization at C-3 in the synthesis of N-N axially chiral compound. The scope of the transformation is explored, and the resulting products are derivatized into polyarenes, with their chiroptical properties analyzed. While the work is suitable for publication, the following revisions are recommended:

Our Response:

We thank the Reviewer for their constructive comments and support for the publication of this work.

Comment 1:

It is suggested to revise the abstract to more accurately reflect prior literature, particularly regarding the synthesis of C–N axially chiral carbazoles. For instance, it is claimed that "chiroptical applications of their atropisomers have remained untapped due to the synthetic challenge of achieving remote atroposelectivity" and that the work "addresses the long-standing synthetic challenge of remote atroposelectivity" should be polished, given that a catalytic method for atroposelective C–N bond formation on carbazoles have been previously reported (ref. 13). A more balanced discussion, acknowledging prior contributions while explaining the new advances in this work, is needed.

Our Response:

We appreciate your incisive suggestions, and we apologize for the overly generalized statements. We have revised the abstract accordingly.

On the abstract paragraph of page 1.

"chiroptical applications of their atropisomers have remained underdeveloped due to the synthetic challenge of achieving remote atroposelectivity necessitated by extended π -systems."
"By establishing an electrostatic steering strategy for remote atroposelectivity, our work paves the way for integrating multifunctional carbazoles into advanced optical and optoelectronic

technologies.”

Comment 2:

Given the structural similarities between the chiral carbazoles reported here and those in prior literature, a more detailed discussion (supported by a graphic in Figure 1) should be included to clearly delineate the new aspects of this study. The authors should explicitly compare their findings with the precedent case (ref. 13), explaining the key differences in methodology.

Our Response:

Again, thank you for advising us. We have changed Figure 1 accordingly by adding Figure 1C.

We also provided further discussions as you have suggested in the third paragraph of page 2 in the revised manuscript.

“Very recently, Huang, Wong, Yeung and coworkers reported organocatalytic asymmetric bromination at C1 of N-aryl carbazoles using chiral phosphoric acids (CPA) and a Lewis base (3-bromo-4-pyrrolylpyridine, Fig. 1C)⁴⁰. Subsequent cross-coupling furnished novel molecules featuring two or three chiral axes that exhibited circularly polarized luminescence (CPL). Nevertheless, highly efficient synthetic methods to access N–C stereogenic carbazoles with differentiated 2,7-bis-substituents for exploiting of the chiroptical functions of this privileged scaffold remain unexplored.”

We hope that by comparing Figure 1C with Figure 1D, the key differences between the two works can be clearly identified. These include the reaction sites (2- and 7-position), the catalyst system (metal catalyzed cross-coupling), the substrates (N–C and N–N stereogenic compounds) and the range of chiroptical properties investigated (CD, CPL, CT-CPL, CP-TADF). We are willing to make additional changes following the Reviewer’s further comments.

Comment 3:

While the coupling reaction is primarily demonstrated at C-2 of carbazole, one example shows C-3 coupling to give N–N axially chiral compound. Can the reaction be applied to C-3 functionalization to form C–N axial chirality?

Our Response:

Thank you for advising us. We have conducted preliminary experiments using substrates designed for C-3 coupling. Although moderate yield (51%) and selectivity (30.5:69.5 er) were obtained, the results demonstrate potential feasibility. Please see the data in SI (page S78).

We have added the results in the main text of the revised manuscript, on the third paragraph of page 10.

“In contrast, 3,6-dichloro *N*-aryl carbazole underwent the desymmetrization reaction in only 69.5:30.5 er (see SI, SI-10).”

Comment 4:

In the transformation from **61** to **62**, the e.r. increases. Is this step a kinetic resolution? A proper discussion is needed. Since **61** is already chiral, an achiral ligand might suffice for its conversion to **62**. The authors should explain why a chiral ligand is necessary in this step and provide additional commentary in the main text to clarify the mechanistic rationale.

Our Response:

We thank the reviewer for the insightful suggestion.

We believe that a kinetic resolution is operating for the reaction from **61** to **62** using the enantiomeric chiral ligand (similar to the reaction from **55** to **59**). We have observed no change in the er if use of achiral ligand in the second step as in the case of converting **64** to **65**.

We have added these points in the main text of the revised manuscript, on the third

paragraph of page 10.

“Subsequent reaction of **61** using the enantiomer of catalyst supported by (*R*)-**L4** yielded product **62** in 93:5:6.5 er. The improvement in enantiopurity is attributed to a secondary kinetic resolution, which favors the major enantiomer of **61** during the second coupling reaction by inducing a configuration opposite to that of the initial step.”

and on the fourth paragraph of page 10.

“Enantio- and site-selective coupling of trichloro-substituted **63** afforded **64** in 97.5:2.5 er (Fig. 5B). Subsequent coupling with excess aryl boronic acid using SPhos as ligand yielded **65** without change in the enantiopurity (97.5:2.5 er).”

N-aryl carbazoles have wide applications in visible-luminescent materials, the chiroptical applications of this kind of organic compounds were seldom reported. In this manuscript, Xing and Zhu described an efficient Pd-catalyzed asymmetric cross-coupling desymmetrization to construct *N*-aryl carbazoles and *N*-pyrrole carbazoles containing C-N and N-N axially chirality with excellent enantioselectivities, and further applied them to circular dichroism(CD), circularly polarized luminescence (CPL), charge-transfer CPL (CT-CPL) and circularly polarized thermally activated delayed fluorescence (CP-TADF), which provide a good solution for classical organic synthesis. The results are valuable and of great interest to readers. The manuscript is well written and the substrate scope has been well investigated. In my opinion, this study is valuable for the publication in *Nat. Commun.* after some minor revisions are carried out.

- (1) How about 1,8-dichoro substituted *N*-Aryl carbazole substrate carboxyl group, for example 1,8-dichloro-9-(3-methyl-[1,1'-biphenyl]-2-yl)-9*H*-carbazole.
- (2) Some remarkable works relate to the constructing of the C-N axially chiral compounds should be referenced.
- (3) How about the racemization energy barrier of the chiral compound **61**? Suggest the author to check the half-life.

Considering that the authors have replied and revised the manuscript according to the reviewers' comments, I recommend the publication of this manuscript in *Nat. Commun.*